# Inhibition of UVB-Induced Inflammation by *Laminaria japonica* Extract via Regulation of nc886-PKR Pathway

**DOI:** 10.3390/nu12071958

**Published:** 2020-06-30

**Authors:** Kwang-Soo Lee, Eunae Cho, Jin Bae Weon, Deokhoon Park, Mathilde Fréchet, Hanane Chajra, Eunsun Jung

**Affiliations:** 1Life Science Institute, BioSpectrum, Yongin 16827, Gyeonggi, Korea; bioyc@biospectrum.com (K.-S.L.); biozr@biospectrum.com (E.C.); biohy@biospectrum.com (J.B.W.); pdh@biospectrum.com (D.P.); 2Clariant Active Ingredients, d’espagne, 31000 Toulouse, France; Mathilde.frechet@clariant.com (M.F.); hanane.chajra@clariant.com (H.C.)

**Keywords:** UVB radiation, nc886-PKR pathway, *Laminaria japonica*, inflammation

## Abstract

Continuous exposure to ultraviolet B (UVB) can cause photodamage of the skin. This photodamage can be inhibited by the overexpression of the non-coding RNA, nc886, via the protein kinase RNA-activated (PKR) pathway. The study aims to identify how UVB inhibits nc886 expression, and it also seeks to determine whether substances that can control nc886 expression can influence UV-induced inflammation, and the mechanisms involved. The results suggest that UVB irradiation accelerates the methylation of the nc886 gene, therefore, reducing its expression. This induces the activation of the PKR, which accelerates the expression of metalloproteinase-9 (MMP-9) and cyclooxygenase (COX-2), and the production of MMP-9, prostaglandin-endoperoxide synthase (PGE_2_), and certain pro-inflammatory cytokines, specifically interleukin-8 (IL-8), and tumor necrosis factor-α (TNF-α). Conversely, in a model of nc886 overexpression, the expression and production of those inflammatory factors are inhibited. In addition, *Laminaria japonica* extract (LJE) protect the levels of nc886 against UVB irradiation then subsequently inhibit the production of UV-induced inflammatory factors through the PKR pathway.

## 1. Introduction

Seaweed has long been used in the East Asian region, including Korea, China, and Japan, for a variety of different purposes, from dietary remedies to culinary flavors, to medicine [1,2,3,4]. While a wide range of brown algae is used, kelp (*Laminaria japonica*) is one of the most popular. Belonging to the order *Laminariales* and the *Laminariaceae* family, kelp (referred to as “Dasima” in Korean, “Haidai” in Chinese, and “Kombu” in Japanese) has long been used in traditional food and medicine across the region. Its versatility is likely from its water-soluble fibers (i.e., alginate and fucoidan) and other fat-soluble ingredients (e.g., fucosterol, fucoxanthin, magnesium, calcium, iron, zinc). Additionally, kelp possesses plenty of polysaccharides that provide anti-inflammatory and antioxidant effects [5,6]. Studies have attempted to use these anti-inflammatory and antioxidant properties to protect skin from the increasing environmental hazards. For example, fucoxanthin is a substance that prevents cell damage generated from oxidative stress [4]. Furthermore, it has been reported that an ethanolic extract of kelp suppresses inflammatory cytokine production in response to lipopolysaccharides (LPS) stimulation [5]. An anti-aging study reported that polysaccharides extracted from kelp prevent the age-related reduction of collagen synthesis in mouse skin [4,7,8,9].

The skin is the first layer of protection against a wide range of environmental hazards, such as ultraviolet (UV) radiation, heat, and infection [6,10]. Most notable of these hazards is ultraviolet B (UVB) radiation, a major external stressor to the skin—one that can cause photoaging, inflammation, and cancer—with its intensity increasing in the face of worsening environmental pollution and ozone depletion [9,11,12]. UVB’s midrange wavelength, between 280 to 320 nm, is absorbed into the epidermis, damaging it by inducing degradation of DNA and RNA and changing cellular signal transduction channels [13]. This radiation also leads to the epigenetic modification of the skin, increasing the risk of skin cancer [14]—one way is through DNA hypermethylation. DNA methylation, a phenomenon that occurs at 5’ cytosine residues in the CpG island, influences gene expressions that are involved in a wide range of biological controls; whereas, DNA hypermethylation is an epigenetic mechanism that silences the expression of tumor suppressor genes [15,16]. It has been reported that DNA methyltransferase (DNMT) leads to methylation at 5’ cytosine residues in the CpG dinucleotide. DNMT1 plays a role in the maintenance of methylation, while DNMT3a and DNMT3b act as de novo methylases. Exposure to UVB, then, causes an increase in the DNMT1, DNMT3a, and DNMT3b, subsequently leading to a change in the patterns of the skin cell’s DNA methylation [17].

It has been found that nc886 (a long non-coding RNA, also known as vtRNA2-1) is a ligand for the protein kinase RNA-activated (PKR), interferon-induced, double-stranded RNA-activated protein kinase, or eukaryotic translation initiation factor 2-alpha kinase 2 (EIF2AK2), and that the nc886-PKR binding inhibits the activation of the PKR [18,19,20]. Allegedly, not only does nc886 have a CpG island, but it is also affected by methylation [15,16]. A number of studies have reported that the expression of nc886 decreases under the influence of methylation, and this impairs the functionality of nc886 as a tumor suppressor [21,22,23,24]. When methylation is activated by UVB and subsequently downregulating nc886 expression, the PKR phosphorylation can be induced in the nc886-PKR pathway [25]. Such activation of PKR also activates the mitogen-activated protein kinases (MAPKs) pathway, which includes c-Jun N-terminal kinases (JNKs) and p38, to induce the production of metalloproteinase-9 (MMP-9) and inflammatory cytokine [26,27,28,29,30,31,32,33].

This study aims to investigate the potential mitigating effect of *Laminaria japonica* extract (LJE) on inflammatory reactions of the skin when applied to UVB-induced nc886-PKR pathways. Additionally, it aims to elucidate the regulatory effect of nc886 on the PKR signal transduction channel induced by UVB, thereby inhibiting photodamages.

## 2. Material and Methods

### 2.1. Extraction of Laminaria japonica

*Laminaria japonica* was purchased from Wando in South Korea. The dried seaweed (10 kg) was suspended in hot water (1 L, 80 ℃) for 3 h and the aqueous extract was reduced to obtain a concentrate. After filtration through activated carbon, the remaining solution was treated with four volumes of ethanol. After centrifugation (5000 rpm for 20 min), the supernatant was concentrated via rotary evaporation under vacuum and lyophilized. Lyophilized extract was dissolved in distilled water at a 1000-fold higher concentration than the final concentration used in media.

### 2.2. Cell Culture, Cell Treatment with LJE and UVB Irradiation, qRT-PCR, and Methylation-Specific PCR

The human keratinocyte (HaCaT) cell line was purchased from the Cell Line Service (Eppelheim, Germany). The cell culture was conducted under controlled conditions at 37 ℃ and 5% CO_2_, with Dulbecco’s modified Eagle’s medium (WELGENE, Gyeongsan, Korea) that contained 10% fetal bovine serum (Thermo Fisher Scientific, Waltham, MA, USA) and 1% of penicillin-streptomycin.

When the cells reached 30% confluence, they were treated with the indicated concentration of LJE and 5-Aza-2’deoxycytidine (Sigma-Aldrich, St. Louis, MO, USA). After 48 h of culture, the media was replaced with the phosphate-buffered saline prior to UVB irradiated to the cells (5 and 10 mJ/cm^2^). These cells were cultured for 24 h and the total RNA was extracted using TRIzol^TM^ (Thermo Fisher Scientific, Waltham, MA, USA). Samples were treated with DNase I (New England Biolabs, Ipswich, MA, USA) to eliminate genomic DNA. The quantitative measurement of RNA was obtained using an Epoch microplate spectrophotometer (BioTek, Winooski, VT, USA). cDNA was synthesized using the amfiRivert cDNA Synthesis Platinum Master Mix (GenDEPOT, Barker, TX, USA). Genomic DNA was extracted using the AccuPrep^®^ Genomic DNA Extraction Kit (BIONEER, Daejeon, Korea). Bisulfite-conversion was conducted using the EZ DNA methylation kit (Zymo Research, Irvine, CA, USA). To measure qRT-PCR and methylation-specific PCR, we used AccuPower^®^ 2X GreenStar™qPCR Master Mix (BIONEER, Daejeon, Korea) and ABI7500 real-time PCR system (Ambion Inc, Austin, TX, USA). The primer sequences are presented below: nc886 forward: 5’-CGGGTCGGAGTTAGCTCAAGCGG-3’; reverse: 5’-AAGGGTCAGTAAGCACCCGCG-3’; MMP-9 forward: 5’-GTACCGCTATGGTTACACTCG-3’; reverse: 5’-GTTTGGAATCTGCCCAGGTC-3’; COX-2; forward: 5’-GATGGAGAGATGTATCCTCCC-3’; reverse: 5’-GCAGCCAGATTGTGGCATAC-3’; 18S rRNA forward: 5’-CGGCTTTGGTGACTCTAGAT-3’; reverse: 5’-GCGACTACCATCGAAAGTTG-3’; GAPDH forward: 5’-CATCAAGAAGGTGGTGAAGCAGG-3’; reverse: 5’-AGTGGTCGTTGAGGGCAATGC-3’; Methyl-nc886; forward: 5’-TTCGGGTCGGAGTTAGTTTAAGCG-3’; reverse: 5’-AATAAACACCCGCGAATCTCG-3’.

### 2.3. DNA and siRNA Transfection for nc886 Overexpression and Knockdown

To obtain DNA for overexpression of nc886 in cells, DNA was amplified with AccuPower^®^ PCR PreMix (BIONEER, Daejeon, Korea) and with having the genomic DNA of the cell HaCaT as the template DNA and nc886 PCR fragment forward: 5’-CTGCTGGACCTAGGTAGACG-3’, reverse: 5’-AATCCATAACGCACTCCGCG-3’, control PCR fragment forward: 5’-CAACCTTGCGTGGCGTGTGAACT-3’, and reverse: 5’-CACATTCACACCTGATTCTGG-3’ as the primer. The amplified DNA was purified using QIAquick PCR Purification Kit (QIAGEN, Hilden, Germany). The purified DNA was transfected with Lipofectamine^TM^ 3000 reagent (Thermo Fisher Scientific, Waltham, MA, USA). To knockdown of nc886, cells were transfected with anti-oligos (si-ctrl and si-nc886) at a concentration of 250 pM by using Lipofectamine™ RNAiMAX reagent (Thermo Fisher Scientific, Waltham, MA, USA). After 24 h of transfection, the UVB irradiation test was performed.

### 2.4. Western Blotting

Protein extraction from cells treated with LJE and UVB irradiation was done by using CytoBuster™ Protein Extraction Reagent (EMD Millipore Corp, Burlington, MA, USA) that contained the Xpert Phosphatase (GenDEPOT, Barker, TX, USA) and Protease Inhibitor (GenDEPOT, Barker, TX, USA). The extracted proteins were separated with NuPAGE™ 4–12% Bis-Tris Protein gels (Thermo Fisher Scientific, Waltham, MA, USA) and transferred to the polyvinylidene fluoride (PVDF). The antibodies used for all the blots were as follows: p38 (phosphorylated) (dilution 1:1000), SAPK/JNK (phosphorylated) (dilution 1:1000), c-Jun (phosphorylated) (dilution 1:1000), ATF-2 (phosphorylated) (dilution 1:1000), p38 (dilution 1:1000), SAPK/JNK (dilution 1:1000), c-Jun (dilution 1:1000), ATF-2 (dilution 1:1000) (Cell Signaling Technology, Danvers, MA, USA), PKR (phosphorylated) (dilution 1:500) (R&D Systems, Minneapolis, MN, USA), PKR (dilution 1:1000) (Thermo Fisher Scientific, Waltham, MA, USA), beta-actin (dilution 1:15,000) (Santa Cruz Biotechnology, Inc., Dallas, TX, USA). The scanning densitometric values of each band were analyzed with Image J and represented the graph as ratio to loading control.

### 2.5. Analysis of MMP-9, PGE_2_, IL-8, and TNF-α by ELISA

After indicated incubation, MMP-9, prostaglandin E_2_ (PGE_2_), IL-8, and TNF-α concentration in the culture supernatant were measured by using enzyme-linked immunosorbent assay (ELISA) kit (R&D Systems, Minneapolis, MN, USA) following the manufacturer’s instructions.

### 2.6. Data Availability

All data generated or analyzed during this study are included in this published article (and its Appendix A).

### 2.7. Statistical Analysis

All results are presented as the mean and standard deviation. Student’s t-test and Two-way ANOVA was performed using GraphPad Prism6 (GraphPad Software, Inc., San Diego, CA, USA). All experiments were performed in triplicate and repeated three times.

## 3. Results

### 3.1. Overexpression of nc886 Reduces the Production of MMP-9 and Inflammatory Cytokines

Based on previous studies, we found that the reduction of the nc886 expression caused by UVB irradiation is associated with the increase of COX-2 and MMP-9 through the PKR pathway [25]. This study verified the role of nc886 on UV-induced inflammatory factors by using the nc886 overexpression model. As a result, the group of keratinocytes with nc886 overexpression showed a significant increase in nc886 expression (Figure 1A). Moreover, MMP-9 and COX-2 gene expression, which were increased by UVB irradiation, were reduced with nc886 overexpression (Figure 1B,C). Additionally, the production of PGE_2_, which is controlled by MMP-9 and COX-2, was reduced (Figure 1D,E). The study also observed whether nc886 can regulate the expression of IL-8 and TNF-α, which share a PKR pathway. Production of IL-8 and TNF-α was suppressed by nc886 overexpression (Figure 1F,G).

### 3.2. UVB Inhibits nc886 Expression by Increasing nc886 Methylation

To decipher how UVB mediates nc886 downregulation in human keratinocytes, a methylation-specific PCR test was used. It has been found that nc886 expression decreases with the methylation of CpG islands. In this regard, the study measured the methylation of nc886 caused by UVB irradiation. As shown in Figure 2A,B, increasing doses of UVB irradiation on the keratinocytes accelerated the reduction of nc886 expression but increased the methylation of nc886. The expression of DNMTs (DNMT1, 3a and 3b) increased under UVB irradiation (Figure 2C). This suggests that the nc886 reduced expression observed, following UVB irradiation, is the consequence of an increase of nc886 DNA methylation.

### 3.3. Laminaria japonica Extract Reduces the Production of MMP-9 and Inflammatory Cytokines

Based on the results above, the study found that increasing nc886 expression, which is inhibited by UVB exposure, has the potential to regulate and inhibit the MMP-9 and inflammatory cytokines (Figure 1). A pre-test was conducted to screen for substances that can increase and restore the nc886 expression suppressed by UVB exposure. The study tested the ability of LJE to inhibit the MMP-9 and inflammatory factors activated by UVB radiation. As shown in Figure 3A, the LJE escalated the expression of nc886 and mitigated the reduction of its expression through exposure to UVB irradiation. This study observed that LJE affected nc886 expression under UVB irradiation. As a result, the LJE suppressed the methylation of the nc886 gene that was increased by UVB (Figure 3A,B). The study also found that LJE modulated MMP-9 and inflammatory factor expressions. *Laminaria japonica* extract decreased gene expression of MMP-9 and COX-2 (Figure 3C,D) and the production of MMP-9, PGE_2_, IL-8, and TNF-α (Figure 3E–H) under basal and UVB conditions.

### 3.4. Laminaria japonica Extract Attenuates the PKR-MAPKs Pathway Activated by UVB

It is widely known that nc886 blocks the PKR pathway and regulates the expression of inflammatory factors. To identify the regulative effect of the nc886-PKR pathway, cells treated with LJE were exposed to UVB, and Western blots were used to study the expression of factors associated with the PKR pathway. The results showed that UVB irradiation accelerated phosphorylation of PKR (Figure 4A,B). Moreover, UVB irradiation accelerated phosphorylation of p38, JNK, c-Jun and ATF-2 (Figure 4C–F). Treatment with LJE suppressed PKR phosphorylation. LJE inhibited the reduction of nc886 caused by UVB irradiation and blocked signal transduction via the PKR pathway to decrease the production of MMP-9 and inflammatory cytokines.

## 4. Discussion

This study determined that the exposure of skin cells to UVB can cause the reduction of nc886 expression, leading to PKR activation. This results in the phosphorylation of the PKR and the induction of MMP-9, PGE_2_ production and the stimulation of inflammatory cytokines [25]. To protect the skin cells from UVB, a relevant strategy could be the stimulation of nc886 expression. In this study, we found that LJE can augment the expression of nc886 to defend the cells exposed to UVB damage. LJE functions as an extra layer of protection against UVB and nc886 regulated the expression of IL-8 and TNF-α increased by UVB exposure (Figure 1) [5,6,34].

Previous studies have explored the role of nc886 in preventing UVB damages using nc886 knockdown and PKR knockout [25]. This study evaluated nc886 overexpression to verify whether it inhibited UVB-induced skin damage. The introduction of nc886 gene containing A and B boxes transcribed by the RNA polymerase III into the cells caused nc886 overexpression (Figure 2A), and the expression of MMP-9 and COX-2 that had been increased by UVB were suppressed (Figure 1B,C). Additionally, nc886 overexpression decreased the production of MMP-9, PGE_2_, IL-8, and TNF-α which were also stimulated by UVB (Figure 1D–G). In contrast, nc886 knockdown suggested that decreasing nc886 resulted in increased expression of MMP-9 and COX-2 genes and increased production of MMP-9, PGE_2_, IL-8, and TNF-α (Appendix A). These results indicate that increased nc886 expression may protect the skin and suppress damages caused by UVB irradiation.

This study showed that the reduction of nc886 was caused by UVB-induced DNA methylation. Several papers have reported that methylation of the nc886 gene inhibits the expression of nc886 and its ability to act as a tumor suppressor [15,16,23]. The result of methylation-specific PCR on nc886 showed that increasing doses of UVB irradiation caused an acceleration of methylation of the nc886 gene (Figure 2B). UVB also increased the expression of DNMTs which are involved in the methylation (Figure 2C). It was also found that the increase of DNMTs through UVB-induced methylation of the nc886 gene, and such augmented methylation resulted in the reduction of the nc886 expression. When treated with Aza-dC to inhibit the action of DNMTs, the methylation of the nc886 gene stimulated by UVB was decreased (Figure 2D). To identify the protective effect of LJE on the skin when extracted in advance, the cells were treated with LJE and irradiated with UVB under the same conditions above. The LJE increased the expression of nc886 under basal and UVB irradiation conditions (Figure 3A). As a result of a change brought about by the LJE in the methylation of the nc886 gene as the cause of increased nc886 expression, the study found that the LJE reduced the level of methylation (Figure 3B). As was in the above results, increased nc886 inhibited expression of MMP-9 and COX-2 (Figure 3C,D) and reduced production of MMP-9, PGE_2_, IL-8, and TNF-α (Figure 3E–H). Such an alleviating mechanism of the LJE was also found with Western blot analysis. UVB suppressed nc886 expression and the activation of the signal transduction channel via activated PKR, stimulating the production of proteins associated with photodamages (Figure 4A–F). *Laminaria japonica* extract limited nc886 DNA methylation caused by UVB irradiation. As a consequence, inhibition of nc886 DNA methylation leads to its transcription and modulation. The results suggested that the application of LJE increased the quantities of long non-coding nc886. *Laminaria japonica* extract-induced promotion of nc886 expression may be a critically important key in developing UVB protective materials.

## Figures and Tables

**Figure 1 nutrients-12-01958-f001:**
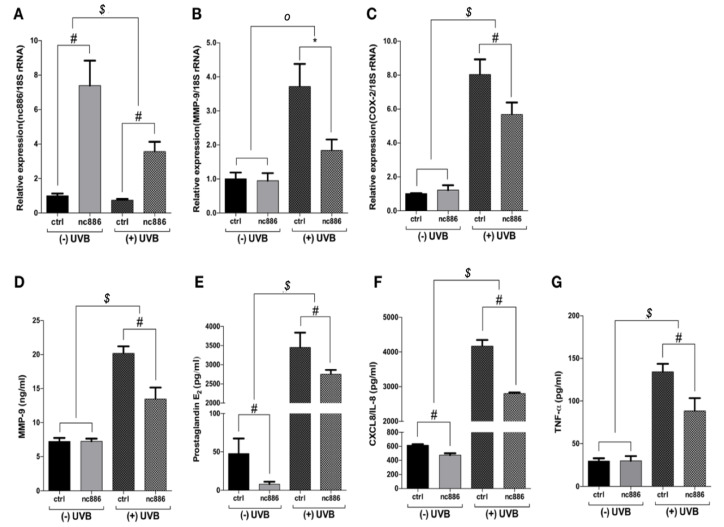
Effect of overexpression nc886 on the production of inflammation-related proteins increased by UVB. Cells were transfected with 1 µg of nc886 gene. Then, 48 h after transfection, the cells were irradiated with UVB (5 mJ/cm^2^). RNA samples were analyzed with quantitative real-time polymerase chain reaction (qRT-PCR) and the cell culture fluid with ELISA. Values measured for (**A**) nc886, (**B**) MMP-9, (**C**) COX-2 were indicated in a fold change compared to the control. The mRNA level was normalized with the expression of 18S rRNA. The measured values for (**D**) MMP-9, (**E**) PGE_2_, (**F**) IL-8, and (**G**) TNF-α were indicated in absolute concentration for each. * *p* < 0.05, # *p* < 0.01, compared to basal levels; *^o^ p* < 0.05, *^$^ p* < 0.01, compared between (−) UVB and (+) UVB groups.

**Figure 2 nutrients-12-01958-f002:**
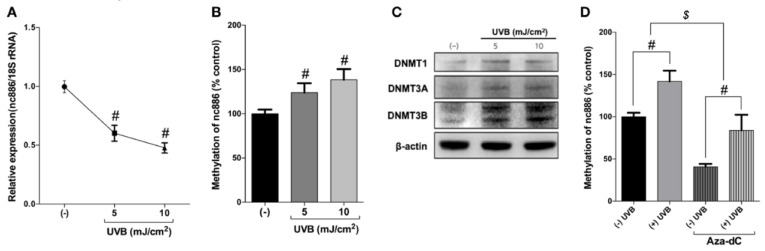
Analysis of the expression and methylation changes of nc886 in human keratinocytes exposed to ultraviolet light UVB. The cells were irradiated with UVB at 5 and 10 mJ/cm^2^. Total RNA and gDNA samples were collected 24 h after UVB irradiation, and protein samples were collected six hours after UVB irradiation. Samples were analyzed by (**A**,**B**,**D**) qRT-PCR, and (**C**) Western blot analysis. The nc886 mRNA level was normalized with 18S rRNA and the methylated-nc886 level was normalized with glyceraldehyde 3-phosphate dehydrogenase (GAPDH). The levels of expression were compared to the control for (**A**) nc886, illustrated in a fold change. A percentage compared to the control was used to indicate the levels for (**B**,**D**) methylation of nc886. *# p* < 0.01, compared to basal levels; *^$^ p* < 0.01, compared between (−) Aza-dC and (+) Aza-dC groups.

**Figure 3 nutrients-12-01958-f003:**
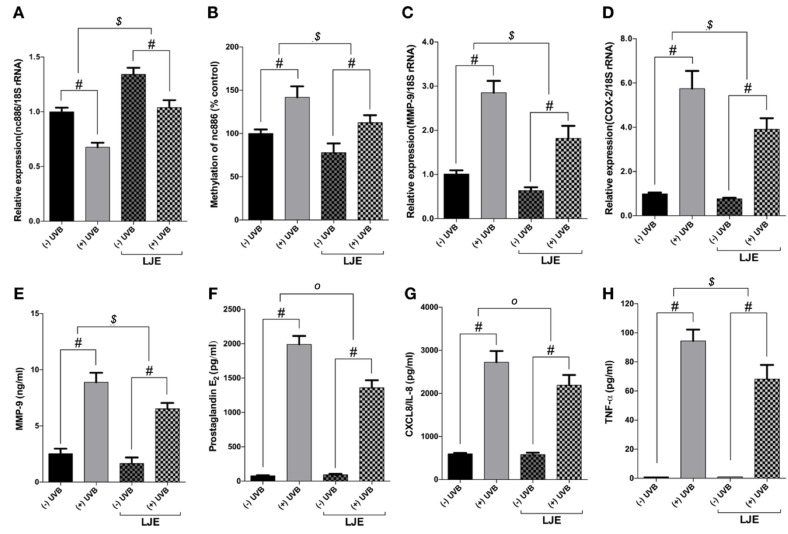
Effect of *Laminaria japonica* extract (LJE) on the production of inflammation-related proteins increased by UVB. The cells were treated with LJE at a concentration of 100 µg/mL for 48 h prior to UVB irradiation (5 mJ/cm^2^). After 24 h of irradiation, RNA, gDNA samples, and cell culture fluid were collected. mRNA and gDNA samples were analyzed with qRT-PCR. A fold change was used to compare nc886 (**A** expression and (**B**) methylation, (**C**) MMP-9, and (**D**) COX-2 to the control. The measured values for (**E**) MMP-9, (**F**) PGE_2_, (**G**) IL-8, and (**H**) TNF-α were indicated in an absolute concentration for each. ** p* < 0.05, *# p* < 0.01, compared to basal levels; *^o^ p <* 0.05, *^$^ p <* 0.01, compared between (−) LJE and (+) LJE groups.

**Figure 4 nutrients-12-01958-f004:**
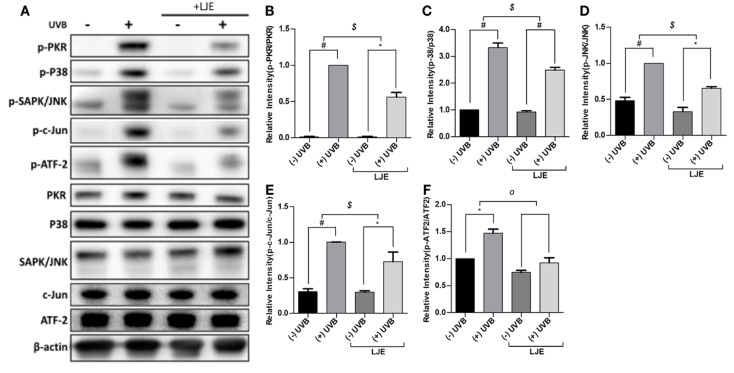
Effect of LJE on UVB-mediated MAPK signaling in keratinocytes. Keratinocytes were treated with LJE (100 µg/mL) for 48 h prior to UVB irradiation (5 mJ/cm^2^). Six hours after irradiation, the PKR phosphorylation and MAPK activation were analyzed with (**A**) Western blots, (**B–F**) the quantitative results are shown. ** p* < 0.05, *# p* < 0.01, compared to basal levels; *^o^ p* < 0.05, *^$^ p <* 0.01, compared between (−) LJE and (+) LJE groups.

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
