# Peer review of "Inhibition of UVB-Induced Inflammation by Laminaria japonica Extract via Regulation of nc886-PKR Pathway"

_nutrients, 2020, doi:10.3390/nu12071958_

Round 1
Reviewer 1 Report
Attached is the pdf manuscript with comments embedded as sticky notes.
The Authors have taken great care to prepare the manuscript.
Probably for future applied research, it would be good to identify the active biomolecules in Laminaria japonica that actually impart these UVB alleviation properties. This would also pave way for commercial usage in the pharmaceutical and cosmetic industries.

Author Response
Thank you for your comments. You have helped us improve the quality of the manuscript.

Reviewer 2 Report
The current manuscript deals with the biological characterization of the Laminaria Japonica extract. I believe that this paper is, properly, the detailed scientific contribution in relation to what has already been expressed in a general way to advertise a commercial cosmetic product. Some data (Figure 1) overlapped with those present in the previous paper (Lee et al. 2019). However, I think that the new data supports the assumptions, but many aspects of the manuscript need to be extensively improved:
- The English language and the text editing need to be profoundly improved. Many sentences are wrongly structured and their meaning is not always clear. The biological language often is inappropriate;
- With regard to the experimental section, the authors should perform an MTT assay to evaluate the cytotoxicity range of the extract, determining the IC50 value and justifying, in this way, the value of the used concentration;
- The authors can not fail to never mention the solvent in which the lyophilized extract was resuspended, neither in the methods section, nor in the captions of the figures;
- Section 2.2 (extract preparation) should be placed before 2.1 section, where the extract is actually used for cell treatment;
- Did the authors verify transfection efficiency? This could justify the very limited increase (<9 fold, low for an overexpression) for nc886;
- Did the authors perform preliminary dose-dependent experiments to verify that 48 h for overexpression and 24 h for silencing were the best timing possible?
- In the western blotting section, the dilution values of all antibodies should be reported;
- The authors must indicate which significance test was used to analyze their results;
- How was a significant densitometric analysis possible using such saturated beta-actin signals?
Author Response
1. The English language and the text editing need to be profoundly improved. Many sentences are wrongly structured and their meaning is not always clear. The biological language often is inappropriate;
→Thanks for the advice. We have made extensive revisions to improve clarity and readability.
2. With regard to the experimental section, the authors should perform an MTT assay to evaluate the cytotoxicity range of the extract, determining the IC50 value and justifying, in this way, the value of the used concentration.
→Cytotoxicity was confirmed prior to treatment with LJE. LJE did not affect any significant change on cell viability at concentrations of 10, 50, and 100 μg/ml. (Figures for questions 2 and 5 were added to the last page of the attached manuscript)
3. The authors can not fail to never mention the solvent in which the lyophilized extract was resuspended, neither in the methods section, nor in the captions of the figures;
→Lyophilized extract dissolved in distilled water at 1000-fold higher concentration of final concentration in medium. We added the solvent information in the method section (Ref.; Line 81).
4. Section 2.2 (extract preparation) should be placed before 2.1 section, where the extract is actually used for cell treatment;
→We modified the order as your suggestion.
5. Did the authors verify transfection efficiency? This could justify the very limited increase (<9 fold, low for an overexpression) for nc886;
→We tested transfection efficiency at dose of 0.1, 0.5, and 1 mg nc886 DNA fragment. If the dose was higher than 1 mg, cytotoxic effect by lipofectamine was observed. We also measured the nc886 expression level at 24 h and 48 h. As shown in graph, transfection condition with 1 mg at 48 h showed higher transfection efficiency compared to other condition. (Figures for questions 2 and 5 were added to the last page of the attached manuscript)
6. Did the authors perform preliminary dose-dependent experiments to verify that 48 h for overexpression and 24 h for silencing were the best timing possible?
→Silencing experiments were performed under the conditions tested in previous study (doi: 10.1016/j.bbrc.2019.01.068).
7. In the western blotting section, the dilution values of all antibodies should be reported;
→Dilution concentration for each antibody is added in the method section (Ref.; Line 120-130).
8. The authors must indicate which significance test was used to analyze their results;
→All results are presented as the mean and standard deviation. Student’s t-test was performed using GraphPad Prism 6 (GraphPad Software, Inc., CA, USA). All experiments were performed in triplicate and repeated three times.
9. How was a significant densitometric analysis possible using such saturated beta-actin signals?
→It is hard to determine whether beta-actin saturated or unsaturated. We thought that β-actin can be used as loading control in western blot analysis. Previous reports also showed the similar pattern and analysis method of western blotting (10.18632/aging.101383). The image was analyzed by measuring densitometric values of proteins as ratios to β-actin used as loading control.

Reviewer 3 Report
This research investigated the potential anti-inflammatory role of Laminaria Japonica via the nc886-PKR pathway using a keratinocyte cell line. The authors did solidify the mechanism of how UVB epigenetically suppresses nc886 expression thereby induces inflammation via activation of the PKR pathway. While the title of the manuscript highlighted the role of LJE, however, the role of LJE was not much focused, mostly just showing expressions of pathway markers. I suggest the authors revise the manuscript with additional mechanistic studies.
1. My biggest question from the research is how LJE reduced the methylation levels of nc886. Authors said ‘LJE limited nc886 DNA methylation caused by UVB irradiation by down-regulating the protein expression of DNMTs.’ (Line 246) in the discussion, but I don’t see any data here. Could authors do an experiment(s) to show the treatment of LJE really increases the expression of DNMTs?
2. Does knockdown of nc886 abolish the anti-inflammatory properties of LJE? If authors can show this, it would greatly strengthen your hypothesis that the shown LJE effects were via nc886.
3. In Fig 2A, I can see a clear nc886 reduction (~50-60%) by UVB (5 mJ/cm2). But I don’t see that in Fig 1A ((-) UVB ctrl vs (+) UVB ctrl). The way of statistical analysis shown is not very straightforward. Could you please clarify it? Regarding this, authors need to state which statistical analysis method was performed in each figure legends, or at least in the Methods.
4. How did authors design Methyl-nc886 primers for methylation-specific PCR? Do they cover the whole CpG island of nc886 gene?
5. Authors cited #17 (Golec et al) to state that UVB increases DNMTs expressions in the introduction (Line 64). But the paper is nothing to do with UVB and DNMTs.
Author Response
This research investigated the potential anti-inflammatory role of Laminaria Japonica via the nc886-PKR pathway using a keratinocyte cell line. The authors did solidify the mechanism of how UVB epigenetically suppresses nc886 expression thereby induces inflammation via activation of the PKR pathway. While the title of the manuscript highlighted the role of LJE, however, the role of LJE was not much focused, mostly just showing expressions of pathway markers. I suggest the authors revise the manuscript with additional mechanistic studies.
→Thank you for your valuable advice. As you mentioned above, this study aims to investigate how UVB inhibits nc886 expression, and whether LJE which regulate nc886 expression can prevent UVB induced inflammation. We missed the mechanism study of LJE on upstream signaling pathway of nc886 expression because we focused on the role of LJE on nc886-PKR pathway. We agreed with your comments and want to add mechanistic results of LJE. But unfortunately, revision need to be submitted within 7 days, so it is hard to meet your suggestion. We will check the role of LJE on upstream signaling pathway of nc886 expression in further study.
1. My biggest question from the research is how LJE reduced the methylation levels of nc886. Authors said ‘LJE limited nc886 DNA methylation caused by UVB irradiation by down-regulating the protein expression of DNMTs.’ (Line 246) in the discussion, but I don’t see any data here. Could authors do an experiment(s) to show the treatment of LJE really increases the expression of DNMTs?
→There was a mistake in expression. DMNTs is a potential regulatory pathway of LJE on nc886 DNA methylation, but we did not check the effect of LJE on expression of DMNTs. We modified the discussion section.
2. Does knockdown of nc886 abolish the anti-inflammatory properties of LJE? If authors can show this, it would greatly strengthen your hypothesis that the shown LJE effects were via nc886.
→It is possible that the anti-inflammatory effect of LJE is abolished in nc886 knockdown model as your suggestion. We will observe the effect of LJE in nc886 knocked down model.
3. In Fig 2A, I can see a clear nc886 reduction (~50-60%) by UVB (5 mJ/cm2). But I don’t see that in Fig 1A ((-) UVB ctrl vs (+) UVB ctrl). The way of statistical analysis shown is not very straightforward. Could you please clarify it? Regarding this, authors need to state which statistical analysis method was performed in each figure legends, or at least in the Methods.
→Fig 1A showed the nc886 reduction (40%) by UVB. The difference between Fig1A and Fig 2A might be derived from transfection condition. Sometimes transfection affects the biological activity of the cells. We added the statistical analysis method in Method section (Ref.; Line 140).
4. How did authors design Methyl-nc886 primers for methylation-specific PCR? Do they cover the whole CpG island of nc886 gene?
→Methyl-nc886 primers was designed based on previous report (doi:10.18632/oncotarget. 2047). It is not cover whole CpG island. It was designed to detect partial portion of 100 bp in length from a CpG island of about 270 bp in length.
5. Authors cited #17 (Golec et al) to state that UVB increases DNMTs expressions in the introduction (Line 58). But the paper is nothing to do with UVB and DNMTs.
→There was a mistake. The reference was corrected (Ref.; Line 314)

Round 2
Reviewer 2 Report
- The English language and the text editing need to be profoundly improved. Many sentences are wrongly structured and their meaning is not always clear. The biological language often is inappropriate;
→Thanks for the advice. We have made extensive revisions to improve clarity and readability.
The authors should correct:
Line 80: …..was dissolved
Lines 87-88: sentence not clear
Line 116: sentence not clear
Line 129: unusual biological language
Line 131-133: unusual biological language
- With regard to the experimental section, the authors should perform an MTT assay to evaluate the cytotoxicity range of the extract, determining the IC50 value and justifying, in this way, the value of the used concentration.
→Cytotoxicity was confirmed prior to treatment with LJE. LJE did not affect any significant change on cell viability at concentrations of 10, 50, and 100 μg/ml. (Figures for questions 2 and 5 were added to the last page of the attached manuscript)
Ok
- The authors can not fail to never mention the solvent in which the lyophilized extract was resuspended, neither in the methods section, nor in the captions of the figures;
→Lyophilized extract dissolved in distilled water at 1000-fold higher concentration of final concentration in medium. We added the solvent information in the method section (Ref.; Line 81).
Ok
- Section 2.2 (extract preparation) should be placed before 2.1 section, where the extract is actually used for cell treatment;
→We modified the order as your suggestion.
Ok
- Did the authors verify transfection efficiency? This could justify the very limited increase (<9 fold, low for an overexpression) for nc886;
→We tested transfection efficiency at dose of 0.1, 0.5, and 1 mg nc886 DNA fragment. If the dose was higher than 1 mg, cytotoxic effect by lipofectamine was observed. We also measured the nc886 expression level at 24 h and 48 h. As shown in graph, transfection condition with 1 mg at 48 h showed higher transfection efficiency compared to other condition. (Figures for questions 2 and 5 were added to the last page of the attached manuscript)
Ok
- Did the authors perform preliminary dose-dependent experiments to verify that 48 h for overexpression and 24 h for silencing were the best timing possible?
→Silencing experiments were performed under the conditions tested in previous study (doi: 10.1016/j.bbrc.2019.01.068).
Ok
- In the western blotting section, the dilution values of all antibodies should be reported;
→Dilution concentration for each antibody is added in the method section (Ref.; Line 120-130).
Ok
- The authors must indicate which significance test was used to analyze their results;
→All results are presented as the mean and standard deviation. Student’s t-test was performed using GraphPad Prism 6 (GraphPad Software, Inc., CA, USA). All experiments were performed in triplicate and repeated three times.
Student's t-test is used to compare two averages, if data follows a normal (Gaussian) distribution. If the number of averages is greater than two (several treatment vs ctrl, figg. 3 and 4), ANOVA should be used, with a post-hoc test. The authors should improve this analysis.
- How was a significant densitometric analysis possible using such saturated beta-actin signals?
→It is hard to determine whether beta-actin saturated or unsaturated. We thought that β-actin can be used as loading control in western blot analysis. Previous reports also showed the similar pattern and analysis method of western blotting (10.18632/aging.101383). The image was analyzed by measuring densitometric values of proteins as ratios to β-actin used as loading control.
I’m not in agreement with the authors. Obviously densitometric values of proteins were expressed as ratios to β-actin, used as loading control, but if its signals were too much saturated, as showed in Figg. 2 and 4, I think it is somewhat difficult to correctly appreciate the differences.
Author Response
1. The English language and the text editing need to be profoundly improved. Many sentences are wrongly structured and their meaning is not always clear. The biological language often is inappropriate;
→ Thanks for the advice. We have made extensive revisions to improve clarity and readability.
The authors should correct:
Line 80: …..was dissolved
Lines 87-88: sentence not clear
Line 116: sentence not clear
Line 129: unusual biological language
Line 131-133: unusual biological language
→ We modified the sentence as your comments.
8. The authors must indicate which significance test was used to analyze their results;
→ All results are presented as the mean and standard deviation. Student’s t-test was performed using GraphPad Prism 6 (GraphPad Software, Inc., CA, USA). All experiments were performed in triplicate and repeated three times.
Student's t-test is used to compare two averages, if data follows a normal (Gaussian) distribution. If the number of averages is greater than two (several treatment vs ctrl, figg. 3 and 4), ANOVA should be used, with a post-hoc test. The authors should improve this analysis.
→ Thanks for the advice. To avoid confusion, we modified the method and figure legend as your suggestion.
9. How was a significant densitometric analysis possible using such saturated beta-actin signals?
→ It is hard to determine whether beta-actin saturated or unsaturated. We thought that β-actin can be used as loading control in western blot analysis. Previous reports also showed the similar pattern and analysis method of western blotting (10.18632/aging.101383). The image was analyzed by measuring densitometric values of proteins as ratios to β-actin used as loading control.
I’m not in agreement with the authors. Obviously densitometric values of proteins were expressed as ratios to β-actin, used as loading control, but if its signals were too much saturated, as showed in Figg. 2 and 4, I think it is somewhat difficult to correctly appreciate the differences.
→ We agree with your opinion. To detect low expression protein, loading quantity was not optimal condition for detecting β-actin, which caused the saturated blot images. In figure 4, non-phosphorylated protein can be used for relative control. We modified the graph as phosphorylated protein/non phosphorylated protein relative to control such as p-PKR/PKR, p-P38/P38, p-SAPK/JNK/SAPK/JNK, p-c-Jun/c-Jun and p-ATF-2/ATF-2.

Reviewer 3 Report
I understand it is hard to do additional experiments. Corrections have been made.
I still have the last suggestion that needs to be corrected, which is the statistical method. The revised manuscript says the student's t-test was used throughout the data. Most of your experimental design has two variables (-,+ UVB / -,+ nc886 or -,+ LJE), where two-way ANOVA should be used. Particularly, I am not sure how you were able to compare '- UVB vs + UVB' in Fig 1, for example, using the student's t-test.
However, I agree the revised manuscript has been improved.
Author Response
I still have the last suggestion that needs to be corrected, which is the statistical method. The revised manuscript says the student's t-test was used throughout the data. Most of your experimental design has two variables (-,+ UVB / -,+ nc886 or -,+ LJE), where two-way ANOVA should be used. Particularly, I am not sure how you were able to compare '- UVB vs + UVB' in Fig 1, for example, using the student's t-test.
However, I agree the revised manuscript has been improved.
→ Thanks for the advice. To avoid confusion, we modified the method and figure legend as your suggestion.